# Human parasagittal dura is a potential neuroimmune interface

Erik Melin[1,2], Geir Ringstad [3,4], Lars Magnus Valnes[5] & Per Kristian Eide [2,5✉]

Parasagittal dura (PSD) is located on both sides of the superior sagittal sinus and harbours arachnoid granulations and lymphatic vessels. Efflux of cerebrospinal fluid (CSF) to human PSD has recently been shown in vivo. Here we obtain PSD volumes from magnetic resonance images in 76 patients under evaluation for CSF disorders and correlate them to age, sex, intracranial volumes, disease category, sleep quality, and intracranial pressure. In two subgroups, we also analyze tracer dynamics and time to peak tracer level in PSD and blood. PSD volume is not explained by any single assessed variable, but tracer level in PSD is strongly associated with tracer in CSF and brain. Furthermore, peak tracer in PSD occurs far later than peak tracer in blood, implying that PSD is no major efflux route for CSF. These observations may indicate that PSD is more relevant as a neuroimmune interface than as a CSF efflux route.

[1] Department of Radiology, Østfold Hospital Trust, Grålum, Norway. [2] Institute of Clinical Medicine, Faculty of Medicine, University of Oslo, Oslo, Norway. [3] Department of Radiology, Oslo University Hospital—Rikshospitalet, Oslo, Norway. [4] Department of Geriatrics and Internal medicine, Sorlandet Hospital, Arendal, Norway. [5] Department of Neurosurgery, Oslo University Hospital—Rikshospitalet, Oslo, Norway. ✉email: p.k.eide@medisin.uio.no

The human brain is immersed in cerebrospinal fluid (CSF) and surrounded by the meninges. How the efflux of CSF takes place has been studied over centuries without any firm conclusion[1,2]. Traditionally, the proposed efflux routes include passage directly to the venous sinuses via arachnoid granulations, along cranial and spinal nerve routes, and across the vessel walls in brain capillaries[1,2].

In 2015, true lymphatic vessels with CSF drainage function were discovered in the dura mater of mice[3,4]. Impairment of these dural lymphatic vessels decreases the clearance of macromolecules and tracers from the brain: In transgenic mice lacking a functional lymphatic system, injected tau clearance was delayed[5]. In wild-type mice where the dural lymphatic vessels were ablated, clearance of injected tracers with different sizes was delayed and, in the same study, ablation of dural lymphatic vessels in transgenic mice models of Alzheimer's disease (AD) led to increased amyloid-beta plaque load in the brain and amyloid-beta depositions in the meninges[6]. These findings highlight the importance of dural lymphatic vessels for clearance from the brain in animal studies and carry implications for several neurodegenerative diseases, including AD.

After intrathecal administration of a CSF tracer in humans, the tracer passed directly from CSF to dura mater adjacent to the superior sagittal sinus, a location where density of lymph vessels is high, the so-called parasagittal dura (PSD)[7]. In previous anatomical studies, PSD also harbours arachnoid granulations and venous channels that were hypothesised to take part in CSF drainage[8]. Moreover, PSD contains fluid vacuoles that may serve as reservoirs for CSF drainage[9].

Based on these observations, constituents of PSD may represent a determining step for clearance capacity of brain solutes via CSF to blood. We, therefore, hypothesise there is an association between the volume of PSD and the clearance function of PSD and that the PSD clearance function can be assessed by time-dependent enrichment of tracer from CSF. Primarily, we asked how various clinical- and imaging variables correlate with PSD volume. Secondarily, we questioned the association between PSD volume and enrichment and clearance of a CSF tracer from various locations, including cerebral cortex, white matter, subarachnoid space, CSF, and PSD. Finally, we compared CSF tracer efflux to PSD with tracer concentrations in blood.

## Results

**Patients with PSD volume segmentations.** The study enrolled 82 patients who underwent intrathecal contrast-enhanced magnetic resonance imaging (MRI) from November 2016 to December 2019. Of these, six patients had to be excluded after imaging due to difficulties to segment PSD for various reasons: three patients with diffuse dural thickening, two with the poor image quality from motion artefacts, and one with cerebral venous sinus thrombosis. The remaining 76 patients were finally included in the material and analysed.

At the Department of neurosurgery, the 76 patients were divided into different categories based on their clinical diagnosis at the time of evaluation for tentative CSF disorders: arachnoid cysts (AC); pineal cysts (PC), idiopathic intracranial hypertension (IIH), spontaneous intracranial hypotension (SIH), hydrocephalus conditions (HC) and dementia (D) (Table 1). Demographic variables of the patients included in the study are presented in Table 1. The category reference (REF) includes patients with no subsequent diagnosis of CSF disorder. For individual data points, see Supplementary Table 1.

Examples of PSD volume segmentations based on fluid-attenuated inversion recovery-MRI (FLAIR-MRI) and 3D images

**Table 1 Patient material.**

| | |
|---|---|
| Total (n) | 76 |
| Age (yrs) | 42.6 ± 15.0 |
| Sex (F/M) | 54/22 |
| BMI (kg/m²) | 28.2 ± 4.6 |
| Disease category | |
| Reference; REF (n) | 23 |
| Arachnoid cyst; AC (n) | 10 |
| Pineal cyst; PC (n) | 11 |
| Idiopathic intracranial hypertension; IIH (n) | 12 |
| Spontaneous intracranial hypotension; SIH (n) | 8 |
| Hydrocephalus conditions; HC (n) | 10 |
| Dementia; D (n) | 2 |

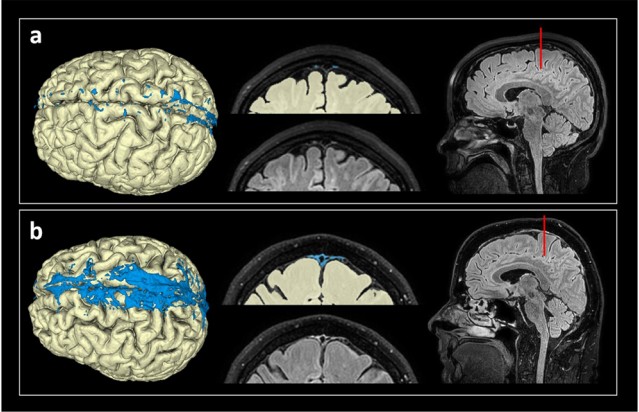

**Fig. 1 3D representations of PSD volume estimation.** The PSD volume (blue) showed marked variability between patients, here illustrated by **a** small PSD volume in a case with IIH and **b** large PSD in a case with SIH.

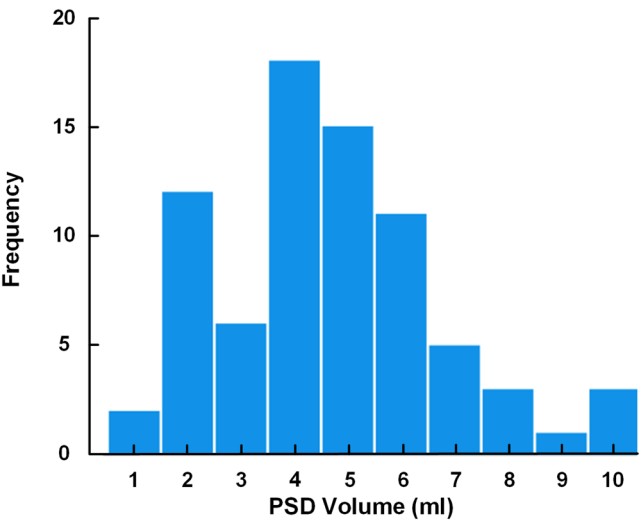

**Fig. 2 Distribution of PSD volume.** In this cohort of $n = 76$ patients, there was a large variation in PSD volume. Individual data points in Supplementary Table 2.

of two different patients are presented in Fig. 1. Considering the entire material, there was a large inter-individual variation in the volume of PSD (Fig. 2), with an average PSD volume of 4.188 ± 2.073 ml (range 0.588 ml to 9.654 ml). Individual data points shown in Supplementary Table 2.

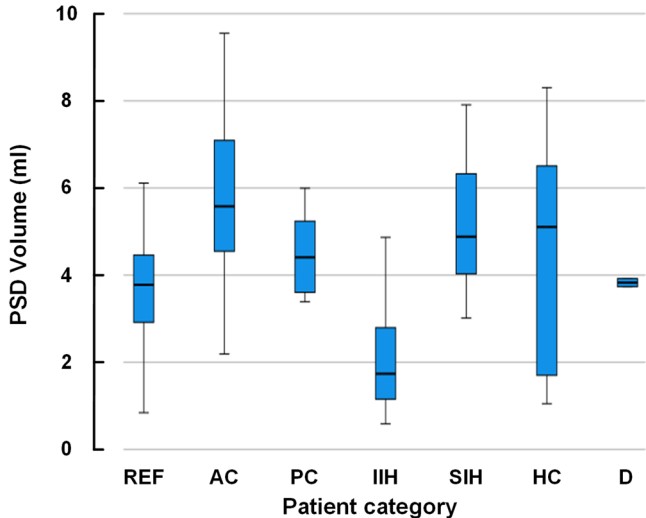

**Fig. 3 Differences in PSD volume between patient categories.** PSD volume differed somewhat between patient categories. REF: Reference, $n = 23$; AC: arachnoid cyst, $n = 10$; PC: pineal cyst, $n = 11$; IIH: idiopathic intracranial hypertension, $n = 12$; SIH: spontaneous intracranial hypotension, $n = 8$; HC: hydrocephalus, $n = 10$; D: dementia, $n = 2$. Data presented as box plots with median, 75% percentiles, and ranges. Individual data points in Supplementary Table 2.

### PSD volume and associations with other variables

*Morphological measures.* The PSD volume correlated with the total intracranial volume ($R = 0.359$, $P = 0.001$), but was not correlated with volume measures of CSF ($R = 0.22$; $P = 0.06$), cerebral cortex grey matter ($R = 0.21$; $P = 0.06$) or subcortical white matter ($R = 0.24$; $P = 0.05$), respectively. Individual data points shown in Supplementary Table 2.

*Demographic variables.* Considering the entire material, there was a positive correlation between age and PSD volume ($R = 0.259$, $P = 0.024$). Moreover, PSD volume was significantly larger in male than female ($5.271 \pm 2.262$ ml vs. $3.747 \pm 1.836$ ml, $P = 0.003$, independent t-test). We found no association between PSD volume and body mass index.

*Diagnosis.* The diagnostic categories differed somewhat regarding PSD volume, with an overall significant difference between groups ($P < 0.001$, ANOVA; Fig. 3). The IIH cohort presented with reduced PSD volume as compared with the SIH cohort ($2.107 \pm 1.306$ ml vs. $5.177 \pm 1.626$ ml, $P = 0.011$; ANOVA with Bonferroni corrected post-hoc tests).

*Sleep.* To explore whether PSD volume is associated with sleep, we examined the correlation between PSD volume and subjective sleep quality (Pittsburgh Sleep Quality Index, PSQI). Considering all patients with PSQI scores ($n = 68$), there was no association between PSD volume and subjective sleep quality ($R = -0.038$, $P = 0.76$). Moreover, poor sleepers (PSQI > 5) and good sleepers (PSQI ≤ 5) did not differ concerning PSD volume. It should be noted, however, that IIH patients presented with poorer sleep quality than REF patients, and that the PSD volume was reduced in IIH versus REF patients.

*Intracranial pressure.* There was a significant negative correlation between PSD volume and the pulsatile intracranial pressure (ICP) as given by the mean ICP wave amplitude (MWA). For the subjects with ICP recordings ($n = 38$), PSD volume correlated negatively with MWA, both regarding the average of MWA

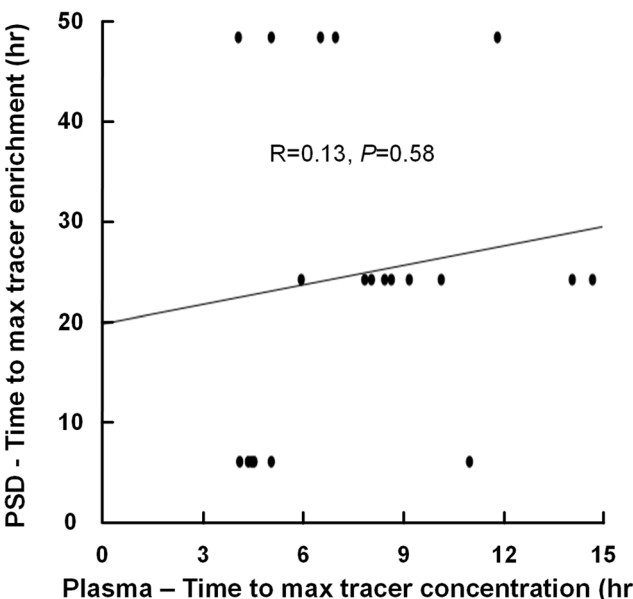

**Fig. 4 Correlation between time to peak of tracer enrichment in PSD and time to maximum plasma concentration in plasma.** The plot shows the correlation between time to peak of tracer enrichment in PSD and time to maximum plasma concentration in plasma for $n = 20$ patients, including the fit line and a non-significant Pearson correlation coefficient. Individual data points in Supplementary Table 3.

($R = -0.395$, $P = 0.014$) and the percentage of recording time with MWA above established the threshold value (>6 mmHg) ($R = -0.441$, $P = 0.009$). The static ICP (mean ICP), on the other hand, was not associated with the PSD volume (average of mean ICP: $R = -0.294$, $P = 0.086$; percentage of time with mean ICP > 20 mmHg: $R = -0.176$, $P = 0.336$).

Multivariable regression analysis was performed to test the strongest predictors of PSD volume. Neither of the variables, age, diagnosis category, or pulsatile ICP (MWA) turned out to have any significant effect on PSD volume. Therefore, none of the presently addressed variables had a strong effect on PSD volume.

### Association between PSD volume and enrichment and clearance of CSF tracer

*CSF tracer in PSD.* There was a strong and highly significant association between tracer enrichment in PSD and tracer enrichment in nearby subarachnoid CSF space at 3 hours ($R = 0.79$, $P < 0.001$), 6 hours ($R = 0.93$, $P < 0.001$), 24 hours ($R = 0.84$, $P < 0.001$) and 48 hours ($R = 0.73$, $P < 0.001$).

Notably, the time to peak tracer enrichment in PSD was not correlated with time to maximum plasma concentration of tracer (Fig. 4; Supplementary Table 3). Furthermore, comparing time to peak tracer enrichment in PSD assessed by MRI with time to maximum plasma concentration of tracer demonstrated significant differences (MRI vs. plasma: $25.636 \pm 15.948$ h vs. $8.113 \pm 3.994$ h, $P < 0.001$; independent samples t-test).

On the other hand, the PSD volume was not strongly associated with the degree of tracer enrichment in PSD. Hence, there was no significant correlation between PSD volume and tracer enrichment after 3 hours ($R = -0.17$, $P = 0.30$), 6 hours ($R = -0.16$, $P = 0.34$) or 24 hours ($R = 0.14$, $P = 0.39$), but there was a positive correlation between PSD volume and tracer enrichment in PSD at 48 hours ($R = 0.398$, $P = 0.010$).

We further noted some differences between patient categories. For example, after 48 hours, tracer enrichment in PSD was

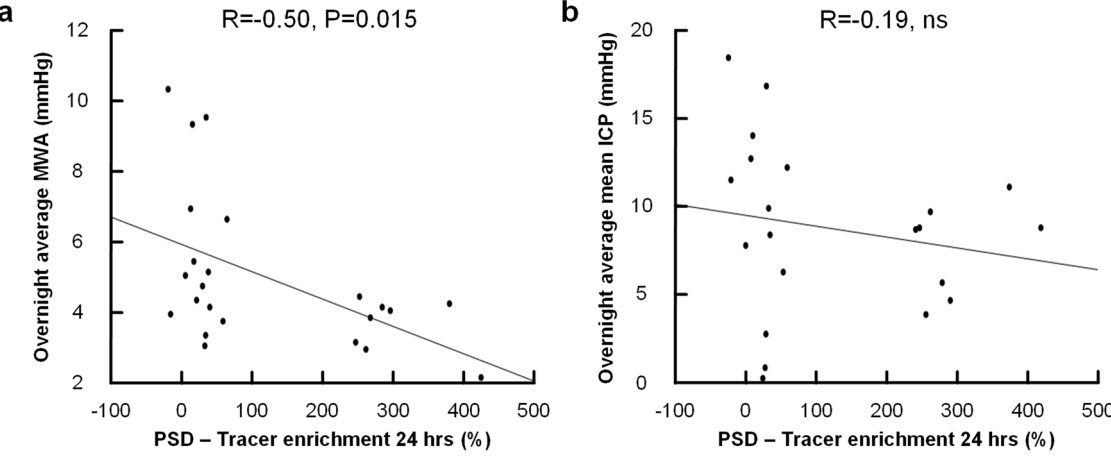

**Fig. 5 The pulsatile and static ICP correlates differently with tracer enrichment in PSD.** The plot shows **a** a significant negative correlation between enrichment in PSD at 24 hours and an overnight average of mean ICP wave amplitude (MWA; pulsatile ICP), but **b** no significant correlation between tracer enrichment in PSD at 24 hours and mean ICP (static ICP). For each plot is shown the fit line and Pearson correlation coefficient for $n = 38$ patients. Individual data points in Supplementary Table 4.

**Table 2 Multivariable regression analysis of variables with strongest impact on PSD tracer enrichment.**

|  | $R^2_{adj}$ | MWA | Tracer in CSF at 3 h |
|---|---|---|---|
| Tracer in PSD at 3 h | 0.729 | −13.0 (95CI −22.3 to −3.8), $P = 0.009$ | 0.27 (95CI 0.19 to 0.35), $P < 0.001$ |
| Tracer in PSD at 6 h | 0.907 | ns | 0.50 (95CI 0.41 to 0.58), $P < 0.001$ |
| Tracer in PSD at 24 h | 0.598 | ns | 0.50 (95CI 0.25 to 0.74), $P < 0.001$ |
| Tracer in PSD at 48 h | 0.568 | ns | 0.55 (95CI 0.27 to 0.83), $P = 0.001$ |

markedly lower in IIH patients than REF subjects (12.6 ± 35.1% vs. 86.8 ± 82.8%, $P = 0.038$).

Furthermore, there was a significant negative correlation between pulsatile ICP (average MWA) and tracer enrichment in PSD at 24 hours ($R = -0.502$, $P = 0.015$; Fig. 5a); This association was not present for the static ICP (mean ICP) ($R = -0-186$, $P = 0.420$; Fig. 5b). With increasing pulsatile ICP, there was also reduced tracer enrichment in CSF ($R = -0.42$, $P = 0.044$), while this was not seen for static ICP ($R = 0.202$, $P = 0.38$). For individual data points, see Supplementary Table 4.

Utilizing multivariable regression analysis, we examined the variables with strongest impact on the tracer enrichment in PSD, which was shown to be tracer enrichment in CSF and the average of pulsatile ICP (MWA) (Table 2). Accordingly, tracer enrichment in the CSF is the variable with the strongest effect on tracer enrichment within the PSD. The effect of pulsatile ICP (MWA) was significant at 3 hours post-injection.

*CSF tracer in cerebral cortex and subcortical white matter.* Tracer enrichment in PSD was significantly correlated with tracer enrichment in the cerebral cortex, subcortical white matter, and the CSF tracer enrichment at vertex (Table 3).

On the other hand, PSD volume was not correlated with tracer enrichment in the various compartments: First, PSD volume did not correlate with tracer enrichment in CSF at 3 hours ($R = -0.31$, $P = 0.06$), 6 hours ($R = -0.18$, $P = 0.28$), 24 hours ($R = -0.006$, $P = 0.97$) or 48 hours ($R = 0.17$, $P = 0.29$). Second, there was no correlation between PSD volume and tracer enrichment in cerebral cortex at 3 hours ($R = -0.12$, $P = 0.35$), 6 hours ($R = -0.14$, $P = 0.25$), 24 hours ($R = -0.04$, $P = 0.73$) or 48 hours ($R = 0.02$, $P = 0.89$). Third, PSD volume was neither correlated with tracer enrichment in subcortical white matter at 3 hours ($R = -0.06$,

$P = 0.65$), 6 hours ($R = -0.12$, $P = 0.33$), 24 hours ($R = -0.02$, $P = 0.87$) or 48 hours ($R = -0.01$, $P = 0.94$). Individual data points of tracer enrichment in the various locations are shown in Supplementary Table 5.

## Discussion

The major observations of this study are that (1) PSD volume shows large inter-individual variations, (2) PSD volume correlates positively with intracranial volume and negatively with intracranial pulse pressure (MWA); however, no single factor explained PSD volume, and neither was it associated with CSF tracer enrichment in brain, CSF or in PSD itself. (3) After intrathecal administration of tracer, time of peak enrichment in CSF around the upper brain convexities and in PSD occurred far later than the occurrence of peak concentration in blood. The dissociation in time between peak enhancement of CSF tracer in PSD and blood, clearly suggests that most CSF tracer was cleared from the subarachnoid space before it reached PSD and that CSF resorption via PSD and dural lymphatic vessels at the vertex are quantitatively less important compared to other efflux routes. This adds to earlier assumptions from cisternography that the subarachnoid space above the cerebral convexities is not a major bulk efflux route[10,11], suggesting CSF efflux from the spinal canal and via cranial nerve outlets at the base of the skull to lymphatic vessels are far more important. Unlike in rodents, we have previously shown that CSF efflux via the cribriform plate to nasal mucosa also seems to be of minor importance[12].

Even if PSD does not seem to constitute a major efflux route, it could still have important functions not reflected by its size or tracer dynamics. It has recently been demonstrated in experimental animal studies that PSD stroma may serve as a major hub

**Table 3 Correlations between PSD tracer enrichment and other locations.**

| Tracer enrichment | Tracer enrichment in PSD | | | | Tracer enrichment in CSF[a] | | | |
|---|---|---|---|---|---|---|---|---|
| | 3 h | 6 h | 24 h | 48 h | 3 h | 6 h | 24 h | 48 h |
| Cerebral cortex | | | | | | | | |
| 3 h | R = 0.27, P = 0.12 | | | | R = 0.44, P = 0.008 | | | |
| 6 h | | R = 0.59, P < 0.001 | | | | R = 0.74, P < 0.001 | | |
| 24 h | | | R = 0.70, P < 0.001 | | | | R = 0.78, P < 0.001 | |
| 48 h | | | | R = 0.61, P < 0.001 | | | | R = 0.67, P < 0.001 |
| Subcortical white matter | | | | | | | | |
| 3 h | R = −0.08, P = 0.63 | | | | R = 0.02, ns | | | |
| 6 h | | R = 0.34, P = 0.05 | | | | R = 0.60, P < 0.001 | | |
| 24 h | | | R = 0.73, P < 0.001 | | | | R = 0.81, P < 0.001 | |
| 48 h | | | | R = 0.63, P < 0.001 | | | | R = 0.68, P < 0.001 |

[a]CSF Subarachnoid CSF at vertex.

for crosstalk between the CNS and peripheral immune system. Here, CNS-derived antigens are presented by dendritic cells to blood-borne T-cells, indicating that the CNS has pushed its immunological boundaries towards its borders[13,14]. In inflammatory and degenerative diseases, T-cells might even enter CSF from PSD[14,15]. How the antigens cross the arachnoid, a tissue by some thought impermeable to macromolecules[2] is not fully understood, but recently two different anatomical studies describe passage of molecules from the subarachnoid space to PSD: The first via endothelial lined channels;[16] The other via arachnoid granulations[17]. Ligation of dural lymphatic vessels in rodents did not reduce passage from CSF to dura, suggesting dural stroma is an intermediate step for CSF efflux before entry into lymph vessels[13]. This experimental finding correlates to our in vivo observation of tracer diffusely spread throughout PSD, far more extensive than only confined to lymph vessels, which probably would be difficult to detect with MRI. In animals, the cribriform plate is a major efflux route for CSF[2], but its importance as an immunological interface has been questioned in favour of dural lymphatics[14]. These observations imply that major CSF efflux routes and CSF immune surveillance could be anatomically separated in both animals and humans.

It should be emphasised that PSD as shown by MRI-FLAIR harbours much more than lymphatic vessels and is therefore not synonymous with lymph. A consistent finding in PSD is intra-dural channels, vacuoles and veins embedded in a stroma[9,18–22], all of that are constituents that may contribute to the high signal in PSD on FLAIR images. The appearance of intradural channels differs somewhat between microscopic studies, but they are not clearly lymphatic[9,21]. Intradural channels have been suggested to function as an intermediate step between CSF and the dural lymph vessels[9]. The CSF tracer observed to diffusely enhance PSD could originate from entry into several of its constituents. One should therefore be cautious to interpret any high signal in PSD on FLAIR images to represent only lymph, as recently done[23]. Our inability to directly depict tubular lymph-like structures within the PSD, even at long-term imaging after injection of CSF tracer, indeed questions whether tubular structures visualised in dura mater shortly after intravenous administration of MRI contrast agent, truly represent lymphatic vessels as previously described[24], but rather contrast within any of the PSD structures including circulating contrast agent within dural veins.

There was a strong correlation between tracer enrichment in the various locations: PSD, CSF, cerebral cortex and subcortical white matter. Being a small molecule, the results indicate that gadobutrol propagation between these compartments is by large governed by diffusion along concentration gradients. Further-more, the findings emphasize the close connection between the interstitial space in the brain, CSF, and PSD and the possibility for fluid exchange along this route with the CSF compartment acting as a bridging link between the brain and the dura.

Both increasing age and male sex correlated positively with PSD volume, consistent with earlier findings[25,26]. Patients with AD had less visible intradural channels compared to younger non-AD individuals in a limited post-mortem study[21]. This could be related to age differences and not AD, but still, an interesting finding as PSD volume increases with age and the channels in PSD decrease with age or AD. The PSD volume was not assessed in that study and the intradural channels could have been col-lapsed or affected by fibrosis[21]. As stated before, FLAIR images will not be able to differentiate the contents of PSD and the increase in volume with age could be attributed to increased connective tissue, lymphatic vasculature, or intradural channels. The small number of patients with dementia in our study ($n = 2$) makes correlations to the various variables, including PSD volume, of limited value for this patient category.

The differences in clearance from the brain and CSF in sleep and wake state have been assessed in both animal and human studies before. In short, clearance of macromolecules from the brain has been shown to increase during sleep[27] and decrease after sleep deprivation[28]. Also, poor sleep quality has been associated with tracer enrichment in regional areas of the brain[29]. A previous animal study has shown increased CSF clearance in wake rodents compared to anesthetized[30], but an in vivo human study of CSF tracer clearance to PSD showed no impact from sleep deprivation[31]. In this study, sleep quality and PSD volume did not correlate. Considering no association between PSD volume and CSF clearance was observed, no correlation between sleep quality and PSD volume is expected.

The differences in PSD volume between patient categories with higher volume in patients with SIH compared to patients with IIH and the negative correlation between PSD volume and pulsatile ICP measurements are noteworthy. There is also a negative correlation between pulsatile ICP and tracer enrichment in PSD and nearby CSF. In a previous study, our group observed increased enrichment and delayed clearance of a CSF tracer from the brain in patients with IIH and increased pulsatile ICP (MWA)[32]. The mechanisms behind SIH and IIH are not known and the differences in PSD volume and tracer dynamics are more probably a result of ICP rather than the causes of the diseases. Reduced compliance, as indicated by increased pulsatile ICP (MWA)[33,34], could lead to altered flow dynamics, both within the brain[32], and in the subarachnoid space[35] with less tracer reaching the convexity of the brain. In addition, the increased pulsatile ICP could also lead to anatomical changes, in this case, decreased volume of PSD.

The CSF tracer was injected at lumbar level, the most used injection site for human in vivo tracer studies throughout history, probably due to a good safety profile. Di Chiro injected radio-iodinated albumin intrathecally at the lumbar level in humans and observed tracer accumulating at the vertex of the skull after 24 hours[36,37]. This could be the result of bulk flow in the cranial direction and main efflux at the vertex as originally described, but our data with peak concentrations in blood before maximum tracer signal in PSD, clearly suggest the tracer has left the subarachnoid space via other routes, and the tracer observed at the vertex in Di Chiros study was a result of limited movement of the parasagittal CSF, as previously discussed by Greitz[10]. Injection to the subarachnoid space intracranially would probably result in a different temporal correlation between maximum signal in PSD and maximum blood concentration, but injection to the intracranial subarachnoid space was not feasible in our study setting.

We utilized correlation analysis in this cohort of tentative CSF disorders for a possible association between PSD volume, disease category, and tracer enrichment. The associations observed do not prove causality, as in the case of differences between the SIH and IIH cohorts. The examined group were heterogeneous with poorly understood clinical conditions of CSF disorders with few patients in each disease category and healthy individuals were not included.

In conclusion, the present results suggest that PSD has no major role in CSF clearance. Its volume could by large neither be explained by other single parameters we investigated, including anatomical- and ICP metrics. Whether the neuroimmune function of PSD is size dependent should be a matter of investigation in further studies.

## Methods

**Ethical approvals.** The study was approved by the Institutional Review Board (2015/1868), the Regional Ethics Committee (2015/96), and the National Medicines Agency (15/04932-7). The patients were included after written and oral informed consent and registered in Oslo University Hospital Research Registry (ePhorte 2015/1868).

**Subjects and study design.** Patients referred to the Department of Neurosurgery, Oslo University Hospital—Rikshospitalet, for CSF circulation disorders were included. Intrathecal contrast-enhanced MRI was done on clinical indication in all subjects. Inclusion criteria were available MRI acquisitions with 3D T1-gradient echo (T1-GRE), 3D T1 Black blood (BB), and 3D T2-Fluid Attenuated Inversion Recovery (FLAIR) sequences. Exclusion criteria were: age <18 years or >80 years; history of hypersensitivity reactions to contrast media agents; severe allergy reactions in general; evidence of renal dysfunction, pregnancy, and breastfeeding. The study design was prospective and observational.

**MRI protocol.** We used a 3 Tesla Philips Ingenia MRI scanner with a 32-channel head coil for the following sequences:

1. 3D T1-gradient echo (T1-GRE): sagittal scanning plane; repetition time (TR) = shortest (typically 5.1 ms); echo time (TE) = shortest (typically 2.3 ms); echo train length (ETL) = 232; flip angle = 8°; 1 average; 1 mm isotropic voxels reconstructed to a 512 × 512 × 368 3D array. Acquisition time: 6 min and 29 s.
2. 3D T1 Black blood (T1-BB): sagittal scanning plane; TR = 700 ms; TE = 35 ms; ETL = 55; flip angle = 80°; 2 averages; 1 mm isotropic voxels reconstructed to a 432 × 432 × 426 3D array. Acquisition time: 4 min and 54 s.
3. 3D T2-Fluid Attenuated Inversion Recovery (FLAIR): sagittal scanning plane; TR = 4800 ms; TE = 311 ms; inversion time = 1650 ms; ETL = 167; flip angle = 90°; 2 averages; 1 mm isotropic voxels reconstructed to a 512 × 512 × 365 3D array. Acquisition time: 5 min and 41 s.

T1-GRE, T1-BB, and FLAIR scans were obtained pre-contrast. T1-GRE and T1-BB were repeated at approximately 0.5, 3, 6, 24, and 48 h after intrathecal injection of 0.5 ml gadobutrol (1 mmol/ml) (Gadovist®, Bayer AB, Sweden). Correct needle position prior to gadobutrol injection was either verified by free CSF flow from the needle or injection of radiopaque iodixanol (Visipaque®, GE Healthcare, Norway). For the first 6 hours after injection the patients were kept in supine position with a pillow under their heads, after that, they could move without any restrictions.

T1-GRE and FLAIR sequences were obtained for all participants. T1-BB images were obtained for 46/76 patients.

**Image post-processing and analysis.** All image processing and analysis were blinded to the patients' age, gender, and clinical data.

For each subject, the T1-GRE volumes for all time points were registered to the same space using FreeSurfer version 6.0 (http://surfer.nmr.mgh.harvard.edu/)[38]. To adjust for intensity differences in the images, the signal intensities between scans were normalized to the signal in the retrobulbar fat at the same time point. This resulted in images of tracer enrichment in the brain over time for each patient. The process of alignment and normalization is more extensively explained in a previous work[39]. The signal intensities for the total cortical grey matter and the white matter were based on these processing steps. The volumes of cortical gray matter and cerebral white matter were obtained from T1-GRE. In five patients the FreeSurfer registration, segmentation, or parcellation failed. No data from dynamic contrast images of the brain were obtained for these patients and all intracranial volumes were instead calculated with SPM12.

Total intracranial and CSF volumes were calculated from T1-GRE using SPM12 and the tissue volumes function included in that software[40]. The results were manually controlled in all subjects and manually corrected if needed.

T1-BB images were used to measure signal intensity in PSD and adjacent CSF. T1-BB images were analysed in the coronal plane in the hospital's picture and archiving system, SECTRA version IDS7 (Sectra AB, Sweden). For each subject and time point, regions of interests (ROI) were placed in PSD and the most adjacent CSF compartment, i.e., in the upper parasagittal subarachnoid space. The signal intensities were normalised to a reference ROI in the vitreous body of the ocular bulb. The data from the ROI measurements in PSD have been published previously in 18/46 patients[7].

We used FLAIR images to segment PSD. PSD was defined as the space with intermediate to high signal next to superior sagittal sinus (SSS) from the most anterior part of the SSS to sinus confluence. A neuroradiologist with 9 years' experience (EM) manually segmented PSD on eight subjects. The first step was to threshold the images in Slicer (version 4.11.20200930) (https://www.slicer.org)[41] to exclude fluid-attenuated CSF and most of the flowing blood. PSD was then manually segmented in three planes using ITK-SNAP version 3.8.0 (http://www.itksnap.org)[42]. Most large veins were excluded in the previous threshold step, but some high-signal vessels had to be removed manually. An in-house artificial intelligence (AI)-software based on a two-dimensional U-Net[43] convolutional network with close resemblance to a tested model, BodySegAI[44], was trained from this data and used to segment PSD in another seven subjects. U-Net was chosen for good performance from a limited training set. The resulting seven AI segmentations were manually corrected by the first author (EM) and used to further train the AI model. The remaining sixty-two subjects were segmented using this latest AI model and manually reviewed and corrected if needed. The AI model was used for more efficient segmentation; Accuracy of the AI model was not estimated.

**ICP measurements.** The overnight ICP measurements included both assessment of the pulsatile and static ICP scores, as previously reported[45,46]. The ICP was

measured using a solid ICP sensor (Codman ICP MicroSensor; Codman, Johnson & Johnson, USA) placed in the brain parenchyma via a small burr hole, enabling online measurements of the static ICP (mean ICP) and pulsatile ICP (mean ICP wave amplitude; MWA) from the same ICP signal[47]. These ICP measurements did not involve CSF drainage or any placement of catheter to the CSF space. The MWA is computed over 6-second time windows and represents the pressure changes taking place during the cardiac cycle. Both the average values of MWA and mean ICP were computed, as well as the percentage of 6-second time windows with MWA ≥ 5 mmHg or mean ICP ≥ 15 mmHg. Notably, all ICP measurements reported here were obtained before interventions. During ICP monitoring, patients were kept in their beds, lying flat during overnight monitoring.

**Sleep quality**. Sleep quality was assessed by a PSQI questionnaire[48] translated to Norwegian[49]. A PSQI-score ≤5 indicates good sleep quality[50]. The patients were asked to answer the questionnaire based on the last months' sleep quality, not only sleep quality for the few days in relation to the MRI exams.

**Plasma concentrations of tracer**. 20/76 patients had plasma concentrations of tracer measured. The plasma concentrations of gadobutrol were estimated from quantifying gadolinium, as previously described in detail[51,52], and her we particularly addressed the time to peak concentration in plasma.

**Statistics and reproducibility**. Continuous data are presented as mean and standard deviation (SD). Normal distribution of the variables was assessed. Continuous data were compared using independent samples t-test. Correlations were determined by Pearson correlation test. The impact of different variables on PSD volume were assessed using multivariable regression analysis. The statistical analysis was performed using SPSS version 27 (IBM Corporation, Armonk, NY). Statistical significance was accepted at the 0.05 level (two-tailed). The method to segment PSD is not a standardised procedure. Two previous studies[25,26] used different protocols to examine and segment PSD and the resulting volumes are not comparable. However, we used the same scanner, coil, protocol, and segmentation method to achieve comparable results within this study, even if the volumes per se are not directly comparable with other studies.

**Reporting summary**. Further information on research design is available in the Nature Portfolio Reporting Summary linked to this article.

## Data availability

The source data are presented in Supplementary File (Supplementary Tables 1–5). Other data analysed in this study are available from the corresponding author on reasonable request.

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

## Acknowledgements

The authors would like to thank Tomas Sakinis, MD, a radiologist at Oslo University Hospital—Rikshospitalet, for providing the AI model used in the segmentations of PSD. Sincerely thanks to the radiologists dr. Øivind Gjertsen, dr. Bård Nedregaard and Dr. Ruth Sletteberg from the Department of Radiology, Oslo University Hospital, who performed the intrathecal injections. The authors also thank the staff at the Intervention Centre and the Department of neurosurgery at Oslo University Hospital—Rikshospitalet, for performing the imaging, and patient care during the examinations. We would like to express gratitude to Are Hugo Pripp, Ph.D., Department of Biostatistics, Epidemiology and Health Economics, Oslo University Hospital, Oslo, for statistical guidance and analyses of the data.

## Author contributions

E.M. Conceptualization, Methodology, Validation, Formal analysis, Investigation Writing—Original Draft, Writing—Review & Editing, Visualization. G.R. Conceptualization, Methodology, Formal analysis, Investigation, Resources Writing—Original Draft, Writing—Review & Editing. L.M.V. Investigation, Writing—Review & Editing. P.K.E. Conceptualization. Methodology, Formal analysis, Resources Writing—Original Draft, Writing—Review & Editing, Visualization, Supervision, Project administration.

## Funding

EM discloses support for the research and publication of this work from South-Eastern Norway Regional Health Authority (Grant number 2020098), and Østfold Hospital Trust (Grant number AB3501). LMV discloses support for the research and publication of this work from South-Eastern Norway Regional Health Authority (Grant number 2020068).

## Competing interests

The authors declare no competing interests.
