## [Peer Review File · Communications Biology]

Reviewers' comments:

Reviewer #1 (Remarks to the Author):

The manuscript by Melin et al describes the CSF flow in parasagittal dura, correlating physiological parameters with various conditions including age, gender, and different diseases in humans. The well-organized manuscript contains a wealth of very accurate measurements and correlations, and a superb discussion. The manuscript settles many open questions and confirms answers to others. My mostly minor criticisms are as follows:

Comments:

1. It would be good to provide an explanation on how intrathecally injected tracer gets into parasagittal dura and how comparable these results are with mouse studies that mainly administer tracers into cisterna magna, lateral ventricle, and brain parenchyma. It is known that the meningeal layers surrounding brain and spinal cord are not identical. Injections into the intrathecal space may not be the best way to mimic CSF movement around brain, including the parasagittal flow and drainage. The authors should address this matter in their response and add it to discussion.
2. The authors should provide more detailed explanation on how patients were divided into different groups; 1) how the patients were chosen for the study; 2) who did the diagnosis and what criteria were used; 3) how patients in different disease groups were divided according to the age/gender/BMI criteria.
3. The n-number in some groups is relatively low (e.g. "dementia" has only 2 patients). Thus, it is impossible to make any clear and robust conclusions on the effects of disease category to PSD function. This should be stated more clearly in the abstract and discussion.
4. Meningeal lymphatic vessels have so far been described to be in dura mater, not in pia or arachnoid mater. Referring to them as meningeal lymphatic vessels has recently caused confusion in the field. Consequently, this reviewer strongly advises changing the term "meningeal" to "dural" throughout the manuscript.
5. There is an ongoing debate about how much the manipulation methods used in brain clearance studies have affected the results presented in various publications. Because of this, the authors could state more accurately what was done in the cited publications.

Minor comments:

- 1) Line 54: please change "meningeal vessels" into "lymphatic vessels".
- 2) Lines 55-56: To the best of my knowledge, the cited publications do not show that dural lymphatic vessels enable the crosstalk between CNS and peripheral immune system via CSF. The authors should either cite the corresponding research articles that show direct evidence or revise the sentences that imply this, accordingly.
- 3) Lines 58-60: The impairment of dural lymphatic vasculature is associated with decreased drainage of macromolecules and cells (and tracers) injected into brain or CSF, not only metabolic solutes, which should be indicated in the sentence. In the paper by Patel et al, K14-sVEGFR3-Ig mice lacking both dorsal and basal dural lymphatic vessels showed reduced clearance of intraparenchymal-injected monomeric Tau from brain. In the paper by Da Mesquita et al, WT mice subjected to verteporfin-mediated photodynamic ablation of dorsal dural lymphatic vessels showed reduced clearance of intraparenchymally injected tracer conjugates of various sizes from brain. Moreover, this paper showed that verteporfin-mediated photodynamic ablation caused more A β accumulation in the brain of transgenic A β producing mice than their littermate controls.
- 4) Lines 174-177: Clearly, discussion of lymphatic vessels as an efflux route should be added!

Reviewer #2 (Remarks to the Author):

Summary statement

This manuscript presented a comprehensive analysis on the parasagittal dura volume and intracranial tracer dynamics. Overall, I find this paper to be well-written and hold obvious significance in the relevant literature. I have a few comments that I hope the authors can kindly address before publication.

Comments:

1. In the introduction, the authors could elaborate more about the potential scientific and clinical value of the presented analysis.
2. In Section Image post-processing and analysis, the authors could provide more information about the reliability of the processing approaches. For example, is there any quantitative evaluation that can be done on the intracranial volumes and the PSD volumes?
3. In Line 335, are there any pre-processing steps for the FLAIR images before input to the AI software? Do the FLAIR images need to be aligned with the T1-GRE images?
4. In Line 335–346, please provide more details on the AI software. Is there a citation for the method? What is the basic network structure for the CNN? How was the accuracy of the segmentation?
5. The intracranial volumes were calculated from T1-GRE while the PSD volumes were calculated from FLAIR images. Will there be different results if both volumes are calculated from FLAIR images?
6. In Line 330 – 334, could the authors provide more explanation about how the T1-BB images were involved in the analysis? For example, which statistical analysis/figures/tables include results from T1-BB images that only acquired for 46 subjects?
7. In Line 369, the authors could provide a brief summary about how the plasma concentrations of tracer were determined.

Manuscript - COMMSBIO-22-3542-T - Response to reviewers with reference to manuscript with changes highlighted

Reviewer #1:

General comment #1:

The manuscript by Melin et al describes the CSF flow in parasagittal dura, correlating physiological parameters with various conditions including age, gender, and different diseases in humans. The well-organized manuscript contains a wealth of very accurate measurements and correlations, and a superb discussion. The manuscript settles many open questions and confirms answers to others. My mostly minor criticisms are as follows:

Answer:

We highly appreciate the reviewer's comments on our manuscript and have answered all comments to the best of our knowledge. Please see our replies and accordingly changes under each specific comment. In our opinion, the manuscript has been much improved.

Specific comment #1:

1. It would be good to provide an explanation on how intrathecally injected tracer gets into parasagittal dura and how comparable these results are with mouse studies that mainly administer tracers into cisterna magna, lateral ventricle, and brain parenchyma. It is known that the meningeal layers surrounding brain and spinal cord are not identical. Injections into the intrathecal space may not be the best way to mimic CSF movement around brain, including the parasagittal flow and drainage. The authors should address this matter in their response and add it to discussion.

Answer:

We thank the reviewer for this insightful comment. Regarding how the tracer reached PSD through the arachnoid membrane, see page 7, paragraph 2, where we also added a recent reference.

The tracer was injected at the lumbar level. Indeed, a portion of the tracer has probably already been absorbed from the intrathecal compartment before the tracer reached the vertex of the subarachnoid space. This would affect the correlation between tracer in PSD and blood. We have added a paragraph under discussion (page 9, paragraph 1).

Specific comment #2:

2. The authors should provide more detailed explanation on how patients were divided into different groups; 1) how the patients were chosen for the study; 2) who did the diagnosis and what criteria were used; 3) how patients in different disease groups were divided according to the age/gender/BMI criteria.

Answer:

Intrathecal contrast-enhanced MRI was done on clinical indication in patients with different types of CSF disorders; their clinical diagnosis refers to their diagnosis in the patient records. This study included patients in whom FLAIR and T1 black-blood sequences were part of the MRI protocol. Inclusion criteria have been defined (page 10, paragraph 1).

Specific comment #3:

3. The n-number in some groups is relatively low (e.g. "dementia" has only 2 patients). Thus, it is impossible to make any clear and robust conclusions on the effects of disease category to PSD function. This should be stated more clearly in the abstract and discussion.

Answer:

We have specified in the abstract that the patients were under work-up for CSF disorders, added to discussion the limited number of dementia patients (page 8, paragraph 2) and in the end of

discussion added “few patients in each disease category” (page 9, paragraph 2). More patients would of course add power to the study, but except for dementia, the n-number are between 8 and 23. The patients were under work-up for CSF disorder, not dementia, and the two patients were only diagnosed with dementia after clinical assessment; the patients were selected for tentative CSF disorder, not dementia.

Specific comment #4:

4. Meningeal lymphatic vessels have so far been described to be in dura mater, not in pia or arachnoid mater. Referring to them as meningeal lymphatic vessels has recently caused confusion in the field. Consequently, this reviewer strongly advises changing the term “meningeal” to “dural” throughout the manuscript.

Answer:

We agree about this important distinction, and have changed the term “meningeal” to “dural”.

Specific comment #5:

5. There is an ongoing debate about how much the manipulation methods used in brain clearance studies have affected the results presented in various publications. Because of this, the authors could state more accurately what was done in the cited publications.

Answer:

We have adjusted the introduction and added details on the studies by Patel et al. and da Mesquita et al. We think this new version is more to the point and more accurate.

Specific comment #6:

Line 54: please change “meningeal vessels” into “lymphatic vessels”.

Answer:

We have corrected this, thank you.

Specific comment #7:

2) Lines 55-56: To the best of my knowledge, the cited publications do not show that dural lymphatic vessels enable the crosstalk between CNS and peripheral immune system via CSF. The authors should either cite the corresponding research articles that show direct evidence or revise the sentences that imply this, accordingly.

Answer:

We have revised the introduction and deleted the sentence that includes references to the immune system. The study’s main aim was to assess PSD volume in patients with tentative CSF disorders and the role of PSD in the immune system is covered in the discussion. We hope you agree the altered introduction is more relevant.

Specific comment #8:

3) Lines 58-60: The impairment of dural lymphatic vasculature is associated with decreased drainage of macromolecules and cells (and tracers) injected into brain or CSF, not only metabolic solutes, which should be indicated in the sentence. In the paper by Patel et al, K14-sVEGFR3-Ig mice lacking both dorsal and basal dural lymphatic vessels showed reduced clearance of intraparenchymal-injected monomeric Tau from brain. In the paper by Da Mesquita et al, WT mice subjected to verteporfin-mediated photodynamic ablation of dorsal dural lymphatic vessels showed reduced clearance of intraparenchymally injected tracer conjugates of various sizes from brain. Moreover, this paper showed that verteporfin-mediated photodynamic ablation caused more A β accumulation in the brain of transgenic A β producing mice than their littermate controls.

Answer:

Please see answer to specific comment #5.

Specific comment #9:

4) Lines 174-177: Clearly, discussion of lymphatic vessels as an efflux route should be added!

Answer:

We have added “to lymphatic vessels” to page 7, paragraph 1. Molecules leaving the intrathecal space from the spinal canal or along nerve routes probably pass to lymphatic vessels, either in lymphatic vessels in the dura or after having passed interstitial tissue outside of the meninges. It is not possible from our MR images to separate lymphatic vessels from other fluid compartments and, besides, it was not the aim of this study.

Reviewer #2:

General comment #1:

Summary statement

This manuscript presented a comprehensive analysis on the parasagittal dura volume and intracranial tracer dynamics. Overall, I find this paper to be well-written and hold obvious significance in the relevant literature. I have a few comments that I hope the authors can kindly address before publication.

Answer:

We highly appreciate the reviewer`s comments and have answered your questions to the best of our knowledge. Please see our replies and referral to subsequent changes in the manuscript under each specific comment.

Specific comment #1:

1. In the introduction, the authors could elaborate more about the potential scientific and clinical value of the presented analysis.

Answer:

We have now adjusted the introduction based on the reviewers` comments. We hope this will make the introduction more relevant (page 3).

Specific comment #2:

2. In Section Image post-processing and analysis, the authors could provide more information about the reliability of the processing approaches. For example, is there any quantitative evaluation that can be done on the intracranial volumes and the PSD volumes?

Answer:

For brain and CSF volumes we used T1-GRE and FreeSurfer [Fischl, B. FreeSurfer. Neuroimage 62, 774-781, doi:10.1016/j.neuroimage.2012.01.021 (2012)]. FreeSurfer failed for 5 patients and instead we used SPM12 and manual correction for intracranial volumes.

For total intracranial volumes, we used T1-GRE and SPM12 instead of FreeSurfer based on this study where SPM12 performed better than FreeSurfer: Malone, I. B. et al. Accurate automatic estimation of total intracranial volume: a nuisance variable with less nuisance. Neuroimage 104, 366-372, doi:10.1016/j.neuroimage.2014.09.034 (2015).

PSD was segmented manually with assistance from an AI-software. A qualitative control was performed for each segmentation and manually corrected if needed. No quantitative evaluation of the results from intracranial volumes and PSD volumes was performed. Further than that, a separate reliability assessment of our image processing approaches would be beyond the scope of the present study.

Specific comment #3:

3. In Line 335, are there any pre-processing steps for the FLAIR images before input to the AI software? Do the FLAIR images need to be aligned with the T1-GRE images?

Answer:

The only pre-processing step used on the FLAIR images before input to the model was z-normalization. The FLAIR images were not aligned with any other images.

Specific comment #4:

4. In Line 335—346, please provide more details on the AI software. Is there a citation for the method? What is the basic network structure for the CNN? How was the accuracy of the segmentation?

Answer:

We added more information about the AI model to the manuscript including citations (page 11, paragraph 4). Accuracy was not estimated, however, all segmentations from the AI model were manually reviewed to assure correctness.

Specific comment #5:

5. The intracranial volumes were calculated from T1-GRE while the PSD volumes were calculated from FLAIR images. Will there be different results if both volumes are calculated from FLAIR images?

Answer:

FreeSurfer and SPM12 are both validated software to segment brain, CSF and total intracranial volumes. T1-GRE is the most used sequence for this purpose. The volume of PSD would probably differ some if different MRI sequences or methods were used, see comment under discussion (page 9, paragraph 3). To our knowledge, segmentations of PSD have only been published two times before, both times with different MRI sequences. We used FLAIR to segment PSD for best contrast to adjacent tissues; T1-GRE is not a good sequence to outline PSD because of poor contrast between PSD and its surroundings.

Specific comment #6:

6. In Line 330 – 334, could the authors provide more explanation about how the T1-BB images were involved in the analysis? For example, which statistical analysis/figures/tables include results from T1-BB images that only acquired for 46 subjects?

Answer:

T1-BB images were used to measure signal intensity in PSD and adjacent CSF. These measurements were used in all analyses involving tracer analyses for PSD including table 2; table 3; Figure 4 and Figure 5. We changed the abstract to clarify that CSF tracer dynamics were not assessed for all 76 patients. CSF tracer analyses for cerebral cortex and subcortical white matter were obtained from T1-GRE and included all patients except for 5 patients where FreeSurfer failed. We have added this to the manuscript (page 11, paragraph 1)

Specific comment #7:

7. In Line 369, the authors could provide a brief summary about how the plasma concentrations of tracer were determined.

Answer:

We have referred to in the text that quantification of gadolinium in plasma to estimate concentrations of gadobutrol in plasma was performed as previously described by our group (page 12, paragraph 3). Since the methodology has been described before, we have not described it in more detail here.

REVIEWERS' COMMENTS:

Reviewer #1 (Remarks to the Author):

The authors have done excellent work with the revision and they have answered my comments thoroughly.

I recommend publication.

Reviewer #2 (Remarks to the Author):

The authors have been responsive to my comments and the paper has been greatly improved. I would recommend this paper for publication.